# Reconstructed Genome-Scale Metabolic Model Characterizes Adaptive Metabolic Flux Changes in Peripheral Blood Mononuclear Cells in Severe COVID-19 Patients

**DOI:** 10.3390/ijms232012400

**Published:** 2022-10-17

**Authors:** Hao Tang, Yanguang Liu, Yao Ruan, Lingqiao Ge, Qingye Zhang

**Affiliations:** Hubei Key Laboratory of Agricultural Bioinformatics, College of Informatics, Huazhong Agricultural University, Wuhan 430070, China

**Keywords:** COVID-19, PBMCs, metabolism, metabolic model, inflammatory, metabolism-directed therapy

## Abstract

Coronavirus disease 2019 (COVID-19) caused by severe acute respiratory syndrome coronavirus-2 (SARS-CoV-2) poses a mortal threat to human health. The elucidation of the relationship between peripheral immune cells and the development of inflammation is essential for revealing the pathogenic mechanism of COVID-19 and developing related antiviral drugs. The immune cell metabolism-targeting therapies exhibit a desirable anti-inflammatory effect in some treatment cases. In this study, based on differentially expressed gene (DEG) analysis, a genome-scale metabolic model (GSMM) was reconstructed by integrating transcriptome data to characterize the adaptive metabolic changes in peripheral blood mononuclear cells (PBMCs) in severe COVID-19 patients. Differential flux analysis revealed that metabolic changes such as enhanced aerobic glycolysis, impaired oxidative phosphorylation, fluctuating biogenesis of lipids, vitamins (folate and retinol), and nucleotides played important roles in the inflammation adaptation of PBMCs. Moreover, the main metabolic enzymes such as the solute carrier (SLC) family 2 member 3 (SLC2A3) and fatty acid synthase (FASN), responsible for the reactions with large differential fluxes, were identified as potential therapeutic targets. Our results revealed the inflammation regulation potentials of partial metabolic reactions with differential fluxes and their metabolites. This study provides a reference for developing potential PBMC metabolism-targeting therapy strategies against COVID-19.

## 1. Introduction

Severe acute respiratory syndrome coronavirus-2 (SARS-CoV-2) poses one of the most momentous threats to human health and safety. As of April 2022, the epidemic of coronavirus disease 2019 (COVID-19) has caused 496 million cases and 6 million deaths worldwide (https://www.who.int/publications/m/item/weekly-epidemiological-update-on-covid-19---12-april-2022 accessed on 12 April 2022). Severe respiratory failure (SRF) frequently occurs in severe cases due to alveolar structure damage in lungs by SARS-CoV-2 [1]. In addition, COVID-19 triggers immune macrophage-activation syndrome [2] and sepsis-induced immunoparalysis, and the latter is characterized by acute lymphopenia, interleukin (IL) -6 (IL-6) accumulation, and the human leukocyte antigen D-related (HLA-DR) exhaustion in CD14 monocytes [1,3]. In addition to the lung injury, SARS-CoV-2 has been reported to mediate the dysfunction of multiple organs by recognizing the surface receptor angiotensin-converting enzyme 2 (ACE2) in central nervous system and pancreatic β cells [4,5].

The accumulation of pro-inflammatory factors is one of the characteristics of COVID-19 [1,6,7,8]. Mechanistically, the replication and release of SARS-CoV-2 destroy lung epithelial cells and exhibit damage-associated molecular patterns, which are recognized by neighboring epithelial cells, endothelial cells, and alveolar macrophages, thereby inducing the production of pro-inflammatory cytokines and chemokines. Subsequently, peripheral monocytes and T cells are recruited by chemokines to the damaged domains to participate in pro-inflammatory processes [9]. Furthermore, the destruction of alveolar structure significantly reduces the arterial partial pressure of blood oxygen (PaO_2_), thus triggering macrophages’ inflammatory response [10]. Excessive pro-inflammatory cytokine accumulation may lead to so-called ‘cytokine storms’, although this phenomenon is not common in severe cases [7]. Undoubtedly, attenuating inflammation is one of dominant strategies to reduce the mortality caused by SARS-CoV-2 infection [11].

Importantly, most subpopulations of peripheral blood mononuclear cells (PBMCs) including monocytes, T cells, and B cells, tend to adaptively respond to cytokine levels in the peripheral circulation or local lesions, and these immune metabolic responses are closely related to cell functions [12,13,14]. Classically, monocytes or macrophages activated by lipopolysaccharide (LPS) or interferon (IFN) -γ (IFN-γ) are polarized from the M 0 immature phenotype to M 1 pro-inflammatory phenotype, which is characterized by aerobic glycolysis [12]. The metabolism of activated T cells mainly consists of aerobic glycolysis and a transition from catabolism to anabolism [13]. The metabolism of T-cell-activated B cells tends to increase the flux of tricarboxylic acid cycle (TCA) and oxidative phosphorylation (OXPHOS) [14]. Adaptive metabolism reconstruction can increase the nutrient uptake and energy production of immune cells, including PBMCs, thus supporting them to exert multiple immune functions, including pro-inflammatory activities. Moreover, reaction substrates or products in the metabolic pathway may be involved in the regulation of immune responses.

Considering this, COVID-19 treatment schemes are expected to be designed from the perspective of immunometabolism. Based on the metabolic profiles of immune cells in response to inflammatory signals, the correlation between the metabolism of inflammatory cells and pro-inflammatory processes has been widely investigated. A number of in vitro studies have indicated that the specific inhibition of metabolic gene expression can change or reverse the phenotype of inflammatory cells and improve anti-inflammatory prognosis by regulating the flux of local metabolic responses. Through the activation of pro-inflammatory cells or the maintenance of pro-inflammatory phenotype, some key proteins indirectly involved in the metabolism can inhibit the development of inflammation by regulating the distribution of metabolic flux. For example, autophagy-related gene 5 (ATG5) has been reported to be a negative regulator of glycolysis and inflammatory cytokine production in dendritic cells infected with respiratory syncytial virus [15]. The crosstalk between NAD-dependent protein sirtuins 1 and energy-sensing enzyme AMP-activated protein kinase (AMPK) supports mitochondrial biogenesis and oxidative metabolism of immune cells, thus promoting anti-inflammatory responses [16]. In order to cope with the rapid mutation of pathogens and avoid the abuse of antibiotics, some immune cell-directed anti-infection strategies have developed [17]. For example, 2-deoxy-D-glucose limits the transcription of human retrovirus HTLV-1 in PBMCs by inhibiting glycolysis [18]. Bedaquiline activates the autophagy and anti-bacterial ability of macrophages via reducing glycolytic flux and enhancing phosphatidylinositol synthesis [19]. The inhibition of the mechanistic target of rapamycin complex 1 (mTORC1) or 3-hydroxy-3-methylglutaryl-coenzyme A (HMG-CoA) reductase reduces cholesterol synthesis in macrophages to accelerate the clearance of engulfed *M. tuberculosis* [20]. Generally speaking, host-directed therapies are less likely to place selective pressure on pathogens, and thus these host-directed therapies, in combination with pathogen-directed drugs, are expected to enhance efficacy. It is necessary to comprehensively investigate the metabolic adaptability (namely, metabolic reprogramming) of PBMCs during inflammation caused by COVID-19 to develop PBMC metabolism-directed therapies.

With the advancement of omics technology, omics analysis has been widely used to characterize the physiology and pathology of specific tissues. However, omics-based analysis alone can no longer meet the requirements of revealing the subtle change in materials and energy under specific conditions. Metabolic networks can comprehensively reflect information on global metabolite circulation and transformation. At present, some algorithms integrate multi-type omics data into genome-scale metabolic models (GSMMs), which makes the reconstruction of context-specific metabolic network more reasonable and convenient. Combined with flux balance analysis (FBA) [21], the reconstructed metabolic network can simulate the global distribution of metabolites and energy flow of cells under specific conditions. This combination is helpful to reveal the mechanism underlying cell metabolic disturbance caused by classical genetic or epigenetic factors. By comparing the flux distribution of the cellular metabolic network between healthy state and disease state, the obtained flux difference nodes may have the potential to become therapeutic targets. Metabolic networks have been widely used to reveal metabolic mechanisms and identify the targets of chronic liver diseases (nonalcoholic fatty liver disease) [22], cancers such as hepatocellular carcinoma [23], and type 2 diabetes mellitus [24].

Metabolic networks can act as platforms to facilitate the studies of viral disease pathogenesis and the screening of related-drug targets, especially for COVID-19. The bulk RNA-Seq/single cell RNA-Seq data from cells infected with SARS-CoV-2 were integrated into human GSMM through the metabolic transformation algorithm rMTA to predict anti-SARS-CoV-2 targets in airway epithelial cells [25]. In addition, the stoichiometric biomass function of SARS-CoV-2 production was integrated into lung cell GSMM to predict the response disturbance that may inhibit virus replication [26]. An algorithm software *findCPcli* was developed for target prediction based on the GSMM of human airway epithelial cells infected with SARS-CoV-2 [27]. The metabolic characterization of PBMC in COVID-19 can be used to predict the severity of infection. A set of PBMC metabolic networks were generated through the iMAT algorithm to predict the metabolic changes in different populations of immune cells against the background of severe SARS-CoV-2 infection [28]. The above studies of context-based metabolic networks contribute to the exploration of the unknown anti-infective capacity of traditional drugs to address the low efficiency problem of drug development. The regulation of metabolic pattern of immune cells may be a direct way to save the severe patients from excessive inflammation since this regulation can reverse the inflammatory phenotype. However, PBMC-based immunotherapy has not been systematically studied in the treatment of COVID-19. Therefore, it is essential to explore the potential therapeutic strategies targeting immunometabolism for acute infection.

In this study, to characterize the metabolism of PBMCs in severe COVID-19 patients, we reconstructed GSMMs by integrating the published single-cell RNA-Seq (scRNA-Seq) data of PBMCs from healthy populations and severe patients using the pyTARG algorithm, and verified the rationality of transcriptome data through traditional gene expression analysis. Our reconstructed GSMM pair was pruned to approach authentic metabolic states. Our results of network flux sampling indicated that the genes responsible for some metabolic reactions with significant flux differences might be potential immunotherapy targets.

## 2. Results and Discussion

### 2.1. Characterization of PBMC Gene Expression Profile in COVID-19

In order to characterize the global transcriptome of PBMC during SARS-CoV-2 infection, the published PBMC scRNA-Seq data were applied to differentially expressed gene (DEG) analysis. The data were obtained from peripheral blood samples from 7 severe patients with similar symptoms and 6 healthy humans. Compared with those from healthy humans, 95% DEGs from severe patients were significantly up-regulated (Appendix A, the number of up- and down-regulated genes were 244 and 13), of which the up-regulated DEGs with the greatest expression difference included multiple immunoglobulin-related genes involved in immune response such as IGLV5-45, IGKV2-30, JCHAIN, and SIGLEC1 (Appendix A). These immunoglobulin-related genes, accounting for the largest proportion (12.8%) of the total up-regulated genes, were essential for humoral immunity. Interferon α-inducible proteins, including IFI27, IFI6, and IFITM3, accounted for the second largest proportion (3.7%). The overexpression of IFITM3 has been reported to significantly inhibit SARS-CoV-2 spike-pseudotyped virus and genuine SARS-CoV-2 infection [29]. S100A8 and S100A9 from the S100 calcium binding protein family, involved in the regulation of various cellular processes, have been identified as biomarkers of SARS-CoV-2 infection, and their increased expression is associated with infection mortality [30,31]. Additionally, some key up-regulated genes are related to SARS-CoV-2 infection (Appendix A). For example, ribonuclease A family member 2 (RNASE2) exhibits anti-viral capacity [32]; the up-regulation of thioredoxin domain containing 5 (TXNDC5) is induced by hypoxia [33]; marginal zone B and B1 cell-specific protein (MZB1) is involved in humoral immunity up-regulation in lung regional lymph nodes of severe COVID-19 patients [34]; lactotransferrin (LTF) displays anti-inflammatory capacity, and competes for binding receptors with SARS-CoV-2 [35]. Such intense activation of immune-related genes is the major response of severe patients to SARS-CoV-2 infection. However, the DEG analysis does not reveal the disturbance of active functional modules, especially metabolic pathways.

Considering this, active functional modules during severe infection were identified via gene set enrichment analysis (GSEA), and GSEA was performed in pre-set gene background with GSEA software. The gene ontology (GO) enrichment analysis indicated that the up-regulated genes were significantly enriched in the GO terms related to immune response (Figure 1A, Appendix A) including “immunoglobulin complex” (GO: 0019814), „sense response to bacterium” (GO: 0042742), “B cell receiver signaling pathway” (GO: 005083). The enrichment trend revealed that PBMCs related to immunity were significantly activated during infection. The increased respiratory activity and energy demand for supporting the extremely active immune activity are the metabolic characteristics of PBMCs during the development of inflammation [36], which was consistent with our results that the enrichment level of genes related to metabolism, especially genes related to respiration or ATP, NADH, and NADPH metabolism, was second only to that of genes related to immune response (Figure 1B, Appendix A). However, the metabolism of PBMCs could not be well characterized due to deficient enrichment metabolic terms and low enrichment scores.

Taken together, the PBMC transcriptome datasets characterized by severe inflammation in COVID-19 patients cannot precisely shape the metabolic profile of PBMCs. Thus, the reconstruction of metabolic networks was implemented.

### 2.2. Metabolic Profile of Reconstructed PBMC Networks

Metabolic network reflects the metabolic disturbance of cells by predicting the reaction rate (or flux) of metabolites. Due to the ambiguous expression differences in most metabolism-related genes, traditional DEG analysis can not reveal the actual metabolic changes of PBMCs. To investigate the metabolic peculiarity of PBMCs during severe infection with COVID-19, a human GSMM covering 5588 metabolites and 8079 reactions for drug development [37,38] was integrated with normalized transcriptome data by constraining the upper or lower boundaries of flux using a pyTARG algorithm [39] (Figure 2). Considering the effects of the tissue specificity of PBMCs and the hypoxia in peripheral circulation during severe infection on metabolic remodeling, some key metabolic reactions were pruned by setting the flux upper/lower boundaries (ub/lb) as zero to simulate the actual metabolic disturbance as much as possible. The modified human metabolic reactions (HMR) included: (1) HMR_8586: ub/lb = 0; since maltose was highly hydrolyzed to glucose by α-glucosidase in the small intestine, it was not directly ingested by PBMCs in blood [40,41]; (2) HMR_3809: ub/lb = 0, since urea cycle of macrophages was incomplete in the absence of ornithine transcarbamylase [42]; (3) HMR_4381, HMR_4375, HMR_4373, HMR_4365, HMR_4103, HMR_4521, HMR_4377: ub = 0; HMR_4368, HMR_4363: lb = 0, because gluconeogenesis mainly occurred in liver and kidney [43,44]; (4) HMR_8743, HMR_4652: lb = 0, because the reversal catalysis of succinate dehydrogenase (SDH) induced by hypoxia resulted in the conversion of fumarate into succinate [45].

After the above HMRs were modified, the context-based GSMM pair representing severe patients and healthy humans was generated. The flux value of each metabolic reaction was predicted by flux sampling, and the flux difference between severe patients and healthy humans was applied to define the regulation state of each reaction. The results showed that extensive elevation of glycolytic fluxes (in HMR_4379, HMR_4373, HMR_4365, HMR_4375, HMR_4363, and HMR_4368) and the serious interception of tricarboxylic acid cycle (in HMR_4139, HMR_8743, and HMR_4652) and electron transport chain (in HMR_6916 and HMR_6918) were basically consistent with aerobic glycolysis of active macrophages and T cells during inflammation [12,13]. Notably, the vigorous purine metabolism mainly containing HMR_4421 was confirmed by one previous metabolomic study of PBMCs in COVID-19 severe patients [46]. The top 25 up-regulated and down-regulated reactions are shown in Table 1 and Table 2.

To determine metabolic modules that were significantly affected during severe COVID-19 infection, reactions with differential fluxes were investigated via hypergeometric test. The enrichment results showed that the transport reaction module, similar to the hubs of material exchange, exhibited the greatest differential fluxes, indicating that this module was most pronouncedly affected. The major reactions with up-regulated fluxes were distributed in “nucleotide metabolism”, “glycolysis”, and “pyrimidine metabolism”, and “purine metabolism” modules (Figure 3A). The reactions with down-regulated fluxes were mainly distributed in “arginine and proline metabolism”, “TCA and glyoxylate/dicarboxylate metabolism”, and “phenylalanine, tyrosine and tryptophanbiosynthesis” modules (Figure 3B). Flux changes in the metabolisms of nucleotides, carbohydrates, and amino acids indicated the metabolic response of PBMCs to COVID-19 infection.

In order to visualize flux differences, metabolic networks were divided into multiple blocks (Figure 4). The glycolysis-TCA-OXPHOS axis was located in the center of metabolic network, and some metabolic modules, such as retinoate metabolism, folate metabolism, pentose phosphate pathway, nucleotide metabolism, and amino acid transports, were clustered or scattered. It should be noted that some key enzymes or metabolites from these modules with flux differences were reported to be related to inflammatory therapies, even COVID-19 treatment. For example, itaconate derivative 4-octyl-itaconate inhibited multiple glycolysis-related enzymes to restrict SARS-CoV-2 replication [47], 25-hydroxycholesterol suppressed the release of multiple viruses by limiting the fusion of virus membrane [48,49], and the folate and retinoate had immunomodulation function [50,51]. Therefore, it could be concluded that the flux distribution of the above multiple modules contributed to the metabolic adaptability of PBMCs in response to inflammation.

Based on metabolic remodeling, we investigated four metabolic module sets with the main flux differences, revealed the relationship between these metabolic changes and immune activation, explored the application prospect of metabolites or targets in these modules against COVID-19. The detailed information on inhibitors, metabolic enzymes, and flux changes in metabolic pathways was summarized in Appendix A.

### 2.3. Aerobic Glycolysis and Lactic Acid Production Are Enhanced in Severe Infection

The metabolic reconstruction results showed that glucose metabolism-related pathways were widely disturbed. Twelve reactions with obviously up-regulated flux were related to glucose transmembrane transport (in HMR_5029 with differential flux of 2.56), glycolysis (in HMR_4379 and 9 other HMRs with differential fluxes of 1.46–5.31), or lactate production (HMR_4388 with differential flux of 2.08). However, the sharp reduction in OXPHOS flux (in HMR_6918 and HMR_6916 with differential fluxes of −1.91 and −2.41) implied that the net production of downstream ATP might have been exhausted. Aerobic glycolysis („Warburg effect”) is a metabolic adaptive transformation with a large consumption of glucose but rapid energy generation, which can occur under aerobic conditions [52]. The above 12 reactions are characterized by glucose metabolism of activated monocytes and CD^4+^ T cells (Th1, Th2 and Th17) during inflammation. The phenotypic polarization M0-to-M1 induced by inflammation leads to the metabolism of macrophages from OXPHOS to aerobic glycolysis, and the latter transforms about 90% of glucose into lactic acid rather than pyruvate [53,54]. The characteristics of aerobic glycolysis included the slow TCA cycle and its flux compensation by the conversion of glutamate into α-ketoglutarate (AKG). Glutamate can compensate for partial carbon cycle loss through glutaminolysis. The exhaustion of glutamine in plasma as a metabolomic feature during COVID-19 infection indicates that adaptive immune cells in peripheral blood may increase the uptake of glutamine [28]. Unexpectedly, our results showed no systematic decrease in TCA cycle flux (in HMR_4456, HMR_4147, HMR_4145, and HMR_4141 with differential fluxes of 1.01–2.54), and aspartate from aspartate-malate shuttle did not compensate fumarate into TCA cycle through the argininiosuccinate cycle (data no shown). The slightly increased flux of the conversion of glutamate into AKG indicated an inappreciable glutamate compensation for TCA cycle (in HMR_3802 and HMR_3807 with a net different flux of 0.06). These unexpected results might be due to immature B cell activation into plasma cells [14]. At the same time, the activated effector CD^8+^ T cells utilized TCA cycle to produce energy [54]. These findings further confirmed that the predicted flux changes were in accordance with the metabolic characteristics related to inflammation in PBMCs.

The enhancement of glycolytic pathway in activated PBMCs was the characteristics of inflammatory response, and enzymes in glycolytic pathway had the potential to be developed into immunometabolic therapy targets [55,56,57,58]. The flux analysis showed that HMR_4373 exhibited the most obvious flux change in the glycolysis pathway (with differential flux of 4.66). Itaconate has been reported to partially reduce the inflammatory response of M1 cells by covalently modifying some cysteines of glycoraldehyde-3-phosphate dehydrogenase (GAPDH, HMR_4373), fructose bisphosphonate aldolase A (ALDOA, HMR_4375 and HMR_4355), and lactate dehydrogenase A (LDHA, HMR_4388), thus downregulating glycolytic flux [59]. The flux analysis also showed that other 3 metabolic reactions (HMR_4375, HMR_4355, and HMR_4388) were also up-regulated to various degrees (Appendix A). In addition, itaconate has also been reported to exert anti-inflammatory function by inhibiting the activity of succinate dehydrogenase (SDH, in HMR_6911 with differential flux of 2.14) and the level of succinic acid [60]. Importantly, dimethyl fumarate and itaconate derivative 4-octyl-itaconate significantly inhibited SARS-CoV-2 replication by activating nuclear factor E2-related factor 2 (Nrf2) [47]. Based on the above findings, it could be concluded that itaconate and its derivatives could be used as candidate drugs for treating COVID-19. Our flux analysis indicated that some metabolic enzymes in the glycolytic pathway of PBMCs were responsible for inflammatory activation, although there was lack of reports on their inhibitors. Multiple activated glycolysis-related metabolic enzymes, such as phosphofructokinase (PFK, in HMR_4379 with differential flux of 5.31), phosphoglycerate mutase (PGAM, in HMR_4365 with differential flux of 4.65), and solute carrier (SLC) family 2 member 3 (SLC2A3) (in HMR_5029 with differential flux of 2.56), are necessary for maintaining macrophage activation [61,62,63]. Therefore, these metabolic enzymes had the potential to act as targets for reducing inflammation and treating COVID-19.

Lactic acidosis, a kind of metabolic acidosis, is a common complication in severe COVID-19 patients, especially gravida, the elderly, or patients suffering from primary diseases. Lactic acidosis mainly results from PaO_2_ decrease, acidic metabolite accumulation, and the viral diarrhea-induced massive loss of bicarbonate [64]. Our flux prediction showed increased lactate accumulation (in HMR_4388 with different flux of 2.08) and release (in HMR_6049 and another seven HMRs with a net differential flux of 5.18) in PBMC during activation. Some inhibitors suppressing lactate accumulation can alleviate or even treat COVID-19. For example, the complement ©-targeting polyclonal antibodies including AMY-101 (C3 inhibitor) [65], eculizumab [66,67] and ravulizumab (both as C5 inhibitors) [68] can restrict the concentration of LDH (HMR_4388) in plasma by reducing complement-mediated tissue injury. Notably, all of the above three drugs targeting C have been studied for the treatment of COVID-19. In addition to inhibiting T cell proliferation, AR-C155858, as an immunosuppressant, can also suppress the monocarboxylate transporter SLC16A1 (HMR_6049 and other 7 HMRs) responsible for lactate transport [69]. Furthermore, stiripentol, an approved LDH inhibitory antiepileptic drug, can inhibit lactate production through combination with LDH in non-competitive manner [70]. The two above-mentioned drugs (AR-C155858 and stiripentol) are promising for treating severe COVID-19 by modulating lactate abundance.

### 2.4. Demand for Lipid Metabolism Increases in Response to Infection

Our reconstructed networks presented the fluctuation in lipid metabolism. The reaction HMR_1917 with extremely high differential flux (5.25) possibly indicated the increased utilization of extracellular cholesterol by PBMCs, although the de novo synthesis of cholesterol was not remarkable. Since cholesterol is involved in the formation of lipid rafts on cell plasma membrane and the synthesis of a variety of biomolecules, the flux surge may contribute to the activation and immune response function of immune cells (T and B cells) [71,72]. Therefore, the inhibition of HMR_1917-related two metabolic enzymes, including Niemann-Pick C intracellular cholesterol transporter 1 (NPC1) and high-density lipoprotein-binding protein (HDLBP), can reduce the inflammatory response induced by COVID-19, although their inhibitors have not been reported. Therefore, NPC1 and HDLBP might be a potential target for the treatment of COVID-19. Interestingly, 25-hydroxycholesterol (25HC), a secondary metabolite of cholesterol (data not shown), can alleviate viral infection by limiting the membrane fusion of viruses (including vesicular stomatitis virus, human immunodeficiency virus (HIV), herpes simplex virus, murine gamma herpesvirus 68) with host cells [48,49].

In addition, the fluctuation in the network was attributed to some fatty acid and sphingolipid metabolism-related reactions. For example, the metabolic pathways of fatty acid synthase (FASN, in HMR_4854, HMR_4855, and HMR_4849 with a differential flux of 1.21–1.57) were up-regulated. FASN can ensure LPS-induced macrophage activation, although fatty acid synthesis-related metabolic pathway flux exhibited no significant differences in this study. Orlistat, a weight-loss drug restraining triglyceride hydrolysis to inhibit FASN, can be used as a potential drug due to its immunomodulatory effect on COVID-19 patients. Lactosylceramide (LacCer), a kind of glycosphingolipids widely distributed in a variety of cells, mainly activates the inflammatory response by stimulating cytoplasmic phospholipase A2 (cPLA2) [73]. Our flux analysis showed that the fluxes of HMR_0786 and another two HMRs related to LacCer biosynthesis were up-regulated to various degrees (with differential fluxes of 1.04–1.32). The increased expression of platelet/endothelial cell adhesion molecule-1 (PECAM-1) promotes the adhesion and recruitment of monocytes to endothelial cells, which might be related to the development of atherosclerosis [74]. Notably, since LacCer mainly activates the inflammatory response by stimulating cPLA2 [73], blocking LacCer cyclic utilization involved in by galactosidase β 1 (GLB1) and glycolipid transfer protein (GLTP) may achieve desirable anti-inflammatory effects against COVID-19.

### 2.5. Pentose Phosphate Pathway Maintains Redox Homeostasis and Nucleotide Metabolism

The activation of the pentose phosphate pathway (PPP) and the intracellular accumulation of reactive oxygen species (ROS) belong to the consequence of PBMC activation [14,36,75,76]. PPP is generally divided into an oxidative phase (oxPPP) and non-oxidative phase PPP (noxPPP). The oxPPP provides a reduction equivalent (such as NADPH) to activated immune cells, which supports the transformation of glutathione from oxidized form (GSSG) to reduced form (GSH). The GSH and thioredoxin are involved in maintaining intracellular redox homeostasis in activated T and B cells [36]. Although the metabolic network cannot directly reflect the change in ROS level in the cytoplasm and mitochondria of PBMCs, there is a frequent conversion between GSSG and GSH of glutathione through HMR_3868, HMR_3869, and HMR_3870 during the activation period, which may indirectly explain the cell response to intracellular ROS accumulation. Regulation of ROS levels is dependent on the PPP, whose upregulation is common during PBMC activation. The regulation of PPP-related active metabolic pathways might partially alleviate the inflammation caused by COVID-19, which remains to be further confirmed in the future studies.

The pentose phosphate of the metabolisms of nucleotide, purine, and pyrimidine are derived from noxPPP. Our results showed that only part of noxPPPs were up-regulated (in HMR_4565 and HMR_4568 with differential flux of 2.50 and 2.03, respectively), and the 36 metabolic reactions (nucleotide, purine, and pyrimidine) with significantly up-regulated fluxes, accounting for ~19% of the total up-regulated reactions, which might be due to noxPPP metabolic adaptation. Although the active purine metabolism has been reported in the metabolomic study of PBMCs in COVID-19 severe patients [46], future studies are suggested to further clarify the relationship between nucleotide, purine, or pyrimidine metabolism and inflammation during COVID-19 infection and to explore the contribution of metabolic disturbance of noxPPP to immune response in PBMCs during severe COVID-19 infection.

### 2.6. Adaptive Biosynthesis Increases of Two Vitamins Folate and Retinoate

Our data showed that the transmembrane transport of folate (FA) (in HMR_7684 with differential flux of 1.54), the transformation of folate into dihydrofolate (DHF) (in HMR_4654 and HMR_4655 with the same differential flux of 1.34), and the transmembrane shuttle of 5-methyltetrahydrofolate (5-MTHF) (in HMR_3916 and HMR_7683 with the same differential flux of 1.59) were significantly increased in PBMCs during severe COVID-19 infection. Folate has been reported to inhibit a wide range of downstream inflammatory responses by limiting monocyte uptake of immunoreactive cyclic dinucleotide [50], and folate transport-related receptor (FR, HMR_7684) is considered as a shortcut for anti-inflammatory drugs to be absorbed by target cells. For instance, methotrexate (MTX), a metabolite with anti-inflammatory and anti-folate capacities, is clinically applied to the treatment of rheumatoid arthritis. The anti-inflammatory effect of its conjugated form (G5-FA-MTX) in combination with FA and synthetic polymer G5 is comparable to G5-MTX [76]. Thus, folate derivatives or FR-related drug design might have the potential to fight against inflammation in COVID-19.

Our metabolic network indicated an increase in the biosynthesis of retinal (in HMR_6630 and HMR_6631 with differential fluxes of 1.52 and 1.23, respectively) and the intracellular accumulation of retinoate (in HMR_6040 and other 4 HMRs with a net differential flux of 0.3). Retinol, retinal, and retinoate (retinoic acid) are all active forms of vitamin A (VA). Retinoate plays an important role in regulating differentiation and functions of macrophages in *trans* or *9-cis* forms [51]. The function impairment of macrophages and lymphocytes is related to the lack of VA in blood [77]. The retinoid has been reported to directly inhibit some viruses’ infections as an immune enhancer, but its relationship with immune metabolism of PBMC remains to be further studied. Retinol is reported to stimulate retinoid-induced gene I (RIG-I) and IFN-α, thus inhibiting infections of measles virus (MeV) and enterovirus 71, respectively [78,79]. AM580, a retinoid derivative and alpha retinoic acid receptor (RAR-α) agonist, can block sterol regulatory element-binding proteins (SREBP)-related lipid synthesis pathway in a RAR-α-independent manner, thus restraining Middle East respiratory syndrome coronavirus (MERS-CoV) replication [80]. Therefore, retinoate and its derivatives are expected to be candidate agents for attenuating COVID-19 infection.

In addition to the above-mentioned four metabolic module sets, more metabolic reactions with remarkable flux changes should be further examined. Two amino acid transport-related proteins SLC7A5 (such as HMR_5473 with differential flux of 7.89) and SLC3A2 (such as HMR_7643 with differential flux of 7.81) have been reported to inhibit inflammation triggered by COVID-19 by activating mTORC1 pathway [81]. The transformation of CO_2_ into HCO^3-^ and the increased efflux of HCO^3-^ (in HMR_3955 and HMR_5448 with differential fluxes of 1.03 and 2.25, respectively) may contribute to alleviate excessive acid accumulation. Carbonic anhydrase 2 (CA2), one of the five cytoplasmic carbonic anhydrases, is abundant in M0, M1, and M2 BMDMs [82]. CA2 inhibitors acetazolamide (ACTZ) and ethoxzolamide (EZA) reduce the secretion of TNF-α and IL-6 in bone marrow-derived M1 cells and inhibit M1-to-M2 polarization, thus achieving anti-inflammatory effect in COVID-19 infection [82]. Our data also showed that the melatonin metabolism pathway (HMR_4548 and HMR_8566 with the same differential flux of 1.53) was significantly up-regulated during severe COVID-19 infection. The broad-spectrum immune regulatory effect of melatonin is controversial in different studies. Melatonin increases IL-2 and TNF-γ to activate human Th1 cells [83], thus contributing to the survival of mouse bone marrow B cells [84]. In terms of mechanism, melatonin enhances glutaminase activity, thus restraining AKG, and eventually inhibiting M0-to-M1 polarization. However, the application of melatonin in anti-inflammation during COVID-19 infection needs to be further investigated.

Furthermore, investigating the expression trend of metabolic enzymes responsible for differential flux can contribute to determining whether these active members possess the potential to become biomarkers of PBMCs for anti-inflammatory treatment. There were 10 up-regulated enzymes responsible for the increased flux of related metabolic reactions, including pyruvate kinase M1/2 (PKM), cytidine/uridine monophosphate kinase 2 (CMPK2), glyceraldehyde-3-phosphate dehydrogenase (GAPDH), SLC38A5, SLC16A1, SLC2A3, glutathione-disulfide reductase (GSR), myeloperoxidase (MPO), cytochrome P450 family 1 subfamily B member 1 (CYP1B1), and galactosidase beta 1 (GLB1). As a known metabolic marker in serum, GSR shows a significant negative correlation with increased IL-10 in COVID-19 patients [85]. In addition, MPO involved in steroid metabolism is considered as a neutrophil-derived infection marker to activate the production of cytokines and ROS in severe SARS-CoV-2 infection [86]. It is necessary to further investigate the abundance of metabolic enzyme in peripheral circulation or PBMCs, which can be applied for correlation analysis with cytokine levels to support supervising the infection process and assessing anti-inflammatory effect.

## 3. Materials and Methods

### 3.1. Analysis of Differentially Expressed Genes (DEGs)

The data used in this study were from the public National Center for Biological Information, GSE150728 (NCBI) database. The peripheral blood mononuclear cell (PBMC) blood samples were collected from 7 severe COVID-19 patients and 6 healthy humans for single-cell RNA-Seq (scRNA-Seq) (accessed on 18 May 2020, https://www.ncbi.nlm.nih.gov/geo/query/acc.cgi?acc=GSE150728) [87]. The donors donating blood samples for the experimental group (admission level of 6/7 patients: ICU) and the control group were distributed in wide age spans (20–80 and 30–50, respectively). Three patients received azithromycin for immunomodulation before being sampled. Five patients were treated with remdesivir after admission. Four patients were treated with remdesivir before being sampled [88]. The RNA-Seq data quality control, read mapping, and gene expression quantification were performed using package “fastp”, “STAR”, and “RSEM”, respectively [88,89,90]. The analysis of DEGs was conducted using “DESeq2” in R package [91], and the analysis results for DEGs are shown in Appendix A and Appendix A.

### 3.2. Gene Ontology Enrichment Analysis of Differentially Expressed Genes

In order to evaluate the expression differences of immune- and metabolic-related DEGs between severe COVID-19 patients and healthy humans, the normalized transcriptome data (transcript per million (TPM)) from the RNA-Seq data were compiled and submitted to the software GSEA_4.1.0 to perform GO enrichment analysis [92]. Gene Ontology gene set database (c5.all.v7.5.1.symbols.gmt) was selected as the enrichment background. The significantly enriched GO terms (|NES| > 1, NOM *p* < 0.05, FDR q < 0.25) and the corresponding normalized enrichment scores (NES) were shown in Figure 1A,B (through R package “circlize”) [93]. All enrichment results are presented in Appendix A.

### 3.3. Genome-Scale Metabolic Model Reconstruction and Flux-Based Reaction Filtration

An updated version of the metabolic model HMR used in this study (MODEL1707250000) was obtained in SBML format from BioModels database [37,38]. Before integrating transcriptomic data, the upper boundaries (ub) or lower boundaries (lb) in the model were set as zero through Python package “cobrapy” to change the direction and availability of specific reactions. The reactions were modified as follows: (1) blocking maltose uptake (HMR_8586: ub/lb = 0) [40,41]; (2) removing ornithine transcarbamylase from macrophages (HMR_3809: ub/lb = 0) [42]; (3) blocking gluconeogenesis (HMR_4381, HMR_4375, HMR_4373, HMR_4365, HMR_4103, HMR_4521, HMR_4377: ub = 0; HMR_4368, HMR_4363: lb = 0) [43,44]; and (4) hypoxia inducing reversal of SDH catalytic direction and converting fumarate into succinate (HMR_8743, HMR_4652: lb = 0) [45]. The integration algorithm pyTARG was employed to restrict the upper and lower boundaries of metabolic reactions based on gene expression and Gene-Protein-Reaction rules (GPR) so as to maximize the consistency between metabolic network and transcriptome background [39]. The average value of gene expression (TPM) of each group (infection group or control group) was used as the input data of integration algorithm.

As objective functions of the PBMC metabolic model cannot be determined, the flux balance analysis (FBA) based on objective function cannot be performed. Thus flux sampling (n = 10,000), rather than FBA, was performed to obtain the metabolic flux distributions of infected and uninfected groups. Artificial centering hit-and-run (ACHR) was selected as the sampler for space sampling in the metabolic model, and the academic version (22.1.0) of IBM ILOG CPLEX was used as the linear programming solver in metabolic modeling. The flux differences in reactions were determined by the following two methods: (1) If the flux directions of a reaction were the same under two conditions (infected vs. uninfected), or if a reaction exhibited no flux (flux=0) under a certain condition, the flux difference was calculated as the difference (infected group—uninfected group); (2) If the two flux directions were the opposite, the flux difference was calculated as the sum of the absolute values of the two groups (|infected group| + |uninfected group|). All sampling results were presented in Appendix A. Based on subsystem classification, the reactions with significant differential fluxes were enriched and analyzed by hypergeometric test, with the results shown in Figure 3 and Appendix A. The flux differences in metabolic modules were visualized by python package “escher” in Figure 4.

## 4. Limitations

In this study, the combination of transcriptomic data and metabolic networks was applied to explain the metabolic perturbations and obtain metabolic targets with differential fluxes in PBMCs of severe COVID-19 patients, so as to expand therapeutic strategies targeting immunometabolism. However, there are some limitations in this study. Firstly, the metabolic reactions with significant differential fluxes were only presented in some subsystems, including purine/pyrimidine metabolism, glycolysis, TCA, transport, etc., and the snapshots of systematic flux difference in most subsystems were not well captured. Secondly, the population composition was complex. The samples of the experimental group were collected from a limited number of male patients with severe diseases who were distributed in a wide agespan and had similar infection symptoms (admission level of 6/7 patients: ICU) [88]. However, in order to comprehensively characterize the physiological diversity of population, more donor samples with a wide agespan and equal sex composition should be included in future research. Next, because of the tissue specificity of PBMC subpopulations, this metabolic network integrating the average gene expression only presented the metabolic average level of various subpopulations. Then, although our pruning behavior shaped the metabolic network closer to tissue specificity, the availability of a highly curated network that systematically reflects the central metabolism of PBMCs should be necessary. Finally, the metabolites or targets with anti-inflammatory or therapeutic potential found in this study need further experimental validation.

## 5. Conclusions

This study integrated the transcriptome data of PBMCs into metabolic network, which contributed to revealing the metabolic characteristics of PBMCs in severe COVID-19 patients. After pruning the inappropriate reactions and integrating transcriptome data, the metabolic model pair was generated to comprehensively reflect monocytes, T cells, and B cells. The active disturbances of some metabolic subsystems responsible for glycolysis, PPP, and biosynthesis of lipid and nucleotide were found to be related to the polarization and activation of PBMCs. Some novel metabolic enzymes, such as SLC2A3, SLC16A1, FASN, and biogenesis-related LacCer, were identified as potential anti-inflammatory targets, and some metabolites, such as folate, retinoate, and melatonin were found to have anti-inflammatory prospects. In addition, some unexpected metabolic changes, including HCO_3_^−^ production, were observed. However, the relationship between these metabolic changes and inflammatory development during COVID-19 infection needs to be further explored. Our findings provide a reference for the development of novel drug targets against COVID-19.

## Figures and Tables

**Figure 1 ijms-23-12400-f001:**
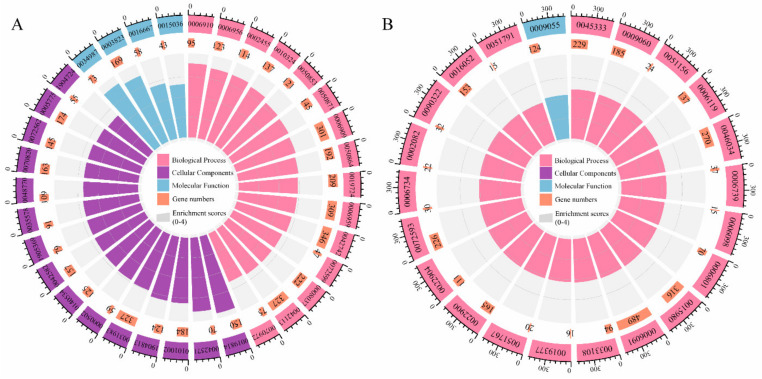
Gene ontology (GO) enrichment analysis of differentially expressed gene (DEGs). Enrichment of up-regulated DEGs related to immune response (**A**) or metabolism (**B**). The three layers of the circle indicate GO terms (outer circles), the number of enriched DEGs (middle circles), and the normalized enrichment scores (NES) indicating the enrichment degree of DEGs at both edges of the target gene sets (inner circles).

**Figure 2 ijms-23-12400-f002:**
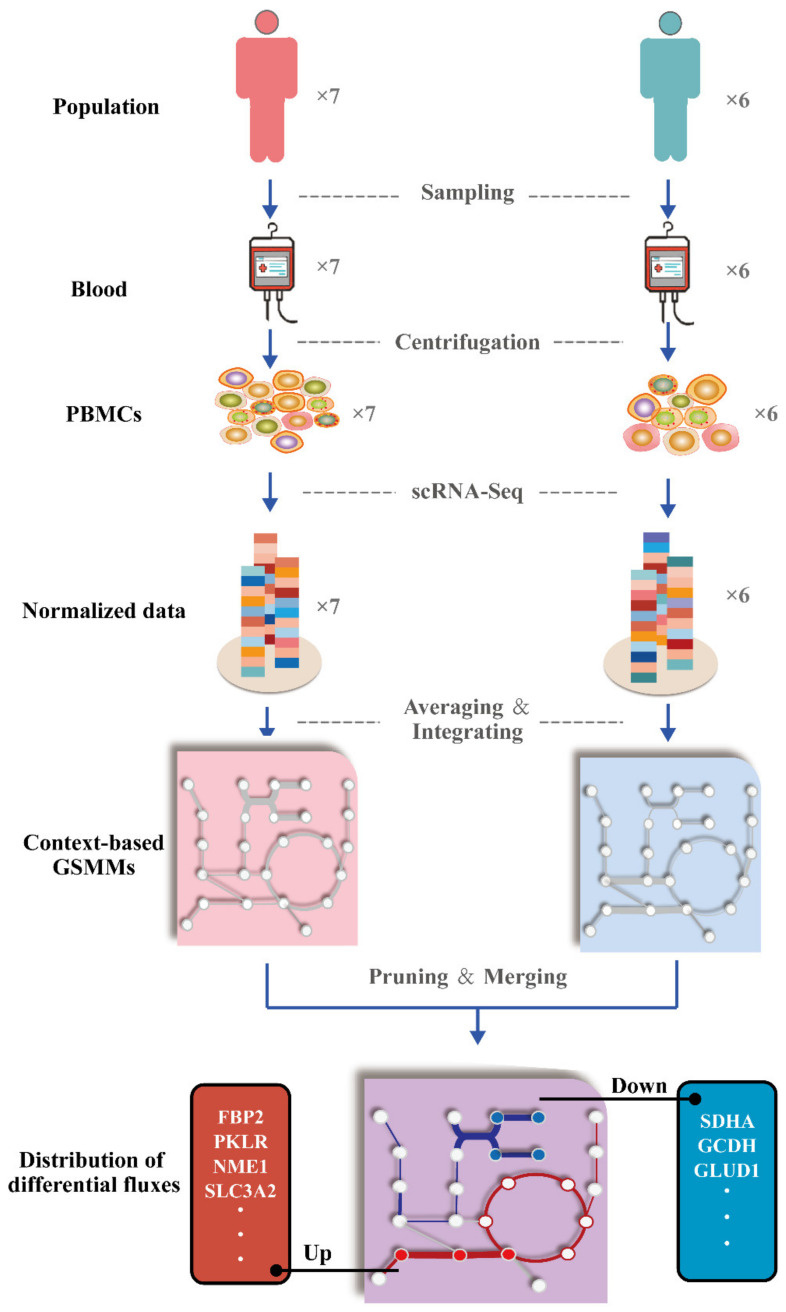
Workflow of the reconstruction of context-based genome-scale metabolic models (GSMMs). The original metabolic network was pruned and integrated with normalized transcriptome data to generate context-based metabolic network pair. Metabolic targets reflecting the differentially metabolic distribution of peripheral blood mononuclear cells (PBMCs) between severe patients and healthy humans were obtained by flux sampling.

**Figure 3 ijms-23-12400-f003:**
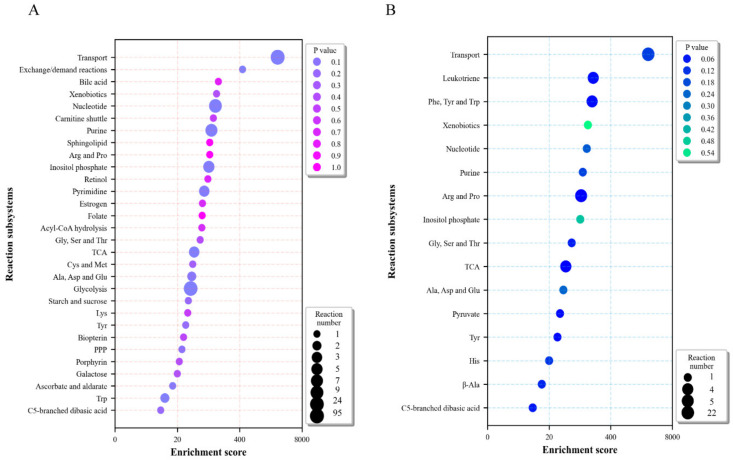
Enrichment of reactions with differential flux. The enrichment of up-regulated (**A**) and down-regulated (**B**) metabolic reactions based on subsystem by hypergeometric test. The bigger the bubble size, the larger the number of reactions enriched in subsystems. The darker the color, the higher the reaction enrichment level in subsystems. *p* < 0.05 was considered as statistically significant. Arg, arginine; Pro, proline; Gly, glycine; Ser, serine; Thr, threonine; Cys, cysteine; Met, methionine; Ala, alanine; Asp, aspartate; Glu, glutamate; Lys, lysine; Tyr, tyrosine; Trp, tryptophan; Phe, phenylalanine; His, histidine; β-Ala, β-alanine; TCA, tricarboxylic acid cycle and glyoxylate/dicarboxylate metabolism; PPP, pentose phosphate pathway.

**Figure 4 ijms-23-12400-f004:**
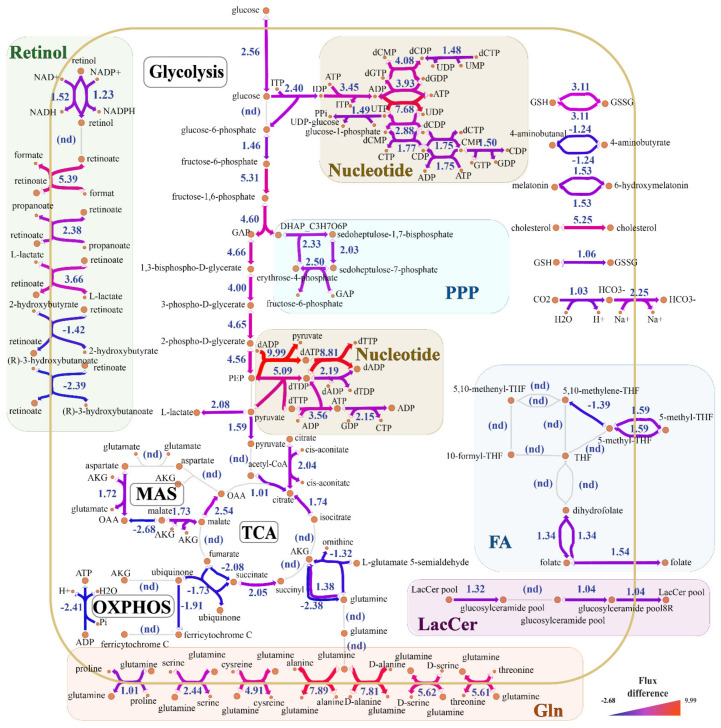
Visualization of flux differences in peripheral blood mononuclear cell (PBMC) metabolic reactions between severe coronavirus disease 2019 (COVID-19) patients and healthy humans. The main differential metabolic pathways were shown in different block colors. The arrow color and thickness indicated the degree of flux difference. PPP, pentose phosphate pathway; Nucleotide, nucleotide/purine/pyrimidine metabolism; MAS, malate-aspartate shuttle; TCA, tricarboxylic acid cycle; OXPHOS, oxidative phosphorylation; FA, folate metabolism; Retinol, retinol metabolism and transport; LacCer, lactosylceramide metabolism and transport; Gln, glutamine transport.

**Table 1 ijms-23-12400-t001:** Top 25 metabolic reactions with up-regulated fluxes between severe coronavirus disease 2019 (COVID-19) patients and healthy humans.

Reaction	Subsystem	Reaction Formula Name	Differential Flux	Reaction Direction
HMR_4421	Purine metabolism	PEP[c] + dADP[c] --> dATP[c] + pyruvate[c]	9.99	Forward
HMR_6614	Pyrimidine metabolism	dTTP[c] + dADP[c] <==> dTDP[c] + dATP[c]	8.81	Reverse
HMR_5473	Transport	glutamine[s] + alanine[c] <==> glutamine[c] + alanine[s]	7.89	Reverse
HMR_7643	Transport	D-alanine[c] + glutamine[s] <==> D-alanine[s] + glutamine[c]	7.81	Reverse
HMR_4006	Pyrimidine metabolism	UTP[c] + ADP[c] <==> UDP[c] + ATP[c]	7.68	Forward
HMR_7644	Transport	D-alanine[c] + glycine[s] <==> D-alanine[s] + glycine[c]	7.42	Forward
HMR_8734	Transport	D-serine[c] + glycine[s] <==> D-serine[s] + glycine[c]	6.68	Reverse
HMR_5458	Transport	alanine[c] + glycine[s] <==> alanine[s] + glycine[c]	5.62	Forward
HMR_8733	Transport	D-serine[c] + glutamine[s] <==> D-serine[s] + glutamine[c]	5.62	Forward
HMR_5497	Transport	threonine[s] + glutamine[c] <==> threonine[c] + glutamine[s]	5.61	Reverse
HMR_6040	Transport	retinoate[s] + formate[c] <==> retinoate[c] + formate[s]	5.39	Forward
HMR_4379	Glycolysis/Gluconeogenesis	fructose-6-phosphate[c] + ATP[c] --> fructose-1,6-bisphosphate[c] + ADP[c]	5.31	Forward
HMR_1917	Transport	cholesterol[c] <==> cholesterol[l]	5.25	Forward
HMR_7885	Nucleotide metabolism	CTP[n] + ADP[n] <==> CDP[n] + ATP[n]	5.15	Forward
HMR_6627	Pyrimidine metabolism	PEP[c] + dTDP[c] <==> dTTP[c] + pyruvate[c]	5.09	Reverse
HMR_5492	Transport	cysteine[c] + glutamine[s] <==> cysteine[s] + glutamine[c]	4.91	Forward
HMR_4373	Glycolysis/Gluconeogenesis	1,3-bisphospho-D-glycerate[c] + NADH[c] + H+[c] <==> GAP[c] + NAD+[c] + Pi[c]	4.66	Reverse
HMR_4365	Glycolysis/Gluconeogenesis	2-phospho-D-glycerate[c] <==> 3-phospho-D-glycerate[c]	4.65	Reverse
HMR_5460	Transport	serine[s] + glycine[c] <==> serine[c] + glycine[s]	4.61	Forward
HMR_4375	Glycolysis/Gluconeogenesis	GAP[c] + DHAP[c] <==> fructose-1,6-bisphosphate[c]	4.60	Reverse
HMR_4363	Glycolysis/Gluconeogenesis	2-phospho-D-glycerate[c] <==> PEP[c] + H2O[c]	4.56	Forward
HMR_8450	Nucleotide metabolism	dGDP[c] + dCDP[c] <==> dGTP[c] + dCMP[c]	4.08	Reverse
HMR_4368	Glycolysis/Gluconeogenesis	1,3-bisphospho-D-glycerate[c] + ADP [c] <==> 3-phospho-D-glycerate[c] + ATP [c]	4.00	Forward
HMR_6041	Transport	(R)-3-hydroxybutanoate[c] + formate[s] <==> (R)-3-hydroxybutanoate[s] + formate[c]	3.99	Forward
HMR_5516	Transport	leucine[c] + methionine[s] <==> leucine[s] + methionine[c]	3.98	Forward

The reactions in “Glycolysis/Gluconeogenesis”, “Transport”, “Purine metabolism”, and “Pyrimidine metabolism” subsystems with major up-regulated fluxes.

**Table 2 ijms-23-12400-t002:** Top 25 metabolic reactions with down-regulated fluxes between severe coronavirus disease 2019 (COVID-19) patients and healthy people.

Reaction	Subsystem	Reaction Formula Name	Differential Flux	Reaction Direction
HMR_7804	Transport	dCDP[c] + dADP[m] <==> dCDP[m] + dADP[c]	−1.08	Reverse
HMR_6006	Transport	2-hydroxybutyrate[c] + acetoacetate[s] <==> 2-hydroxybutyrate[s] + acetoacetate[c]	−1.10	Forward
HMR_4863	Transport	succinate[c] + sulfate[m] <==> succinate[m] + sulfate[c]	−1.13	Reverse
HMR_7808	Transport	dTDP[c] + dUDP[m] <==> dUDP[c] + dTDP[m]	−1.13	Reverse
HMR_4687	Tyrosine metabolism	phenylacetate[m] + NADPH[m] + H+[m] <==> phenylacetaldehyde[m] + NADP+[m] + H2O[m]	−1.51	Forward
HMR_4143	Central carbon metabolism	pyruvate[m] + HCO3-[m] + ATP[m] + H+[m] --> OAA[m] + Pi[m] + ADP[m]	−1.51	Forward
HMR_5470	Transport	threonine[s] + glycine[c] <==> threonine[c] + glycine[s]	−1.52	Reverse
HMR_7892	Nucleotide metabolism	dATP[n] + ADP[n] <==> dADP[n] + ATP[n]	−1.63	Reverse
HMR_6004	Transport	acetoacetate[s] + L-lactate[c] <==> acetoacetate[c] + L-lactate[s]	−1.66	Forward
HMR_4776	Arginine and proline metabolism	AKG[c] + proline[c] + O2[c] --> trans-4-hydroxy-L-proline[c] + succinate[c] + CO2[c]	−1.70	-
HMR_4652	Tricarboxylic acid cycle	fumarate[m] + ubiquinol (UQH2) [m] <==> ubiquinone (UQ)[m] + succinate[m]	−1.73	-
HMR_6025	Transport	acetoacetate[s] + AKG[c] <==> acetoacetate[c] + AKG[s]	−1.74	Reverse
HMR_8464	Nucleotide metabolism	CDP[n] + GDP[n] <==> GTP[n] + CMP[n]	−1.76	Forward
HMR_4787	Arginine and proline metabolism	4-hydroxy-2-oxoglutarate[m] --> glyoxalate[m] + pyruvate[m]	−1.89	-
HMR_6918	Electron transport chain	2 ferricytochrome C[m] + ubiquinol[m] + 2 H+[m] --> 2 ferrocytochrome C[m] + ubiquinone[m] + 4 H+[c]	−1.91	Forward
HMR_8743	Tricarboxylic acid cycle	fumarate[m] + FADH2[m] <==> succinate[m] + FAD[m]	−2.08	-
HMR_5092	Transport	isoleucine[c] <==> isoleucine[s]	−2.12	Reverse
HMR_4785	Arginine and proline metabolism	L-1-pyrroline-3-hydroxy-5-carboxylate[m] + NADP+[m] + 2 H2O[m] --> L-erythro-4-hydroxyglutamate[m] + NADPH[m] + H+[m]	−2.23	Forward
HMR_4242	Aromatic amino acid metabolism	glutaryl-CoA[m] + ubiquinone[m] --> crotonyl-CoA[m] + ubiquinol[m] + CO2[m]	−2.29	Reverse
HMR_3804	Alanine aspartate and glutamate metabolism	AKG[m] + NH3[m] + NADPH[m] + H+[m] <==> glutamate[m] + NADP+[m] + H2O[m]	−2.38	Reverse
HMR_6053	Transport	(R)-3-hydroxybutanoate[c] + retinoate[s] <==> (R)-3-hydroxybutanoate[s] + retinoate[c]	−2.39	Forward
HMR_6916	Electron transport chain	Pi[m] + ADP[m] + 4 H+[c] --> ATP[m] + 4 H+[m] + H2O[m]	−2.41	Forward
HMR_4243	Phenylalanine, tyrosine and tryptophan biosynthesis	glutaryl-CoA[m] + FAD[m] --> crotonyl-CoA[m] + FADH2[m] + CO2[m]	−2.59	Reverse
HMR_4139	Tricarboxylic acid cycle	OAA[c] + NADH[c] + H+[c] <==> malate[c] + NAD+[c]	−2.68	Forward
HMR_5580	Transport	histidine[c] + lysine[s] <==> histidine[s] + lysine[c]	−2.78	Forward

The reactions in “Tricarboxylic acid cycle”, “Transport”, and some amino acids metabolism subsystems with major down-regulated fluxes.

## Data Availability

The data that support the findings of this study are available from the corresponding author upon reasonable request.

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
