# Peer review of "Reconstructed Genome-Scale Metabolic Model Characterizes Adaptive Metabolic Flux Changes in Peripheral Blood Mononuclear Cells in Severe COVID-19 Patients"

_ijms, 2022, doi:10.3390/ijms232012400_

Round 1
Reviewer 1 Report
COVID-19 infection causes metabolic changes in peripheral blood mononuclear cells. In this study, a genome-scale metabolic model was created from differentially expressed gene analysis, which serves as a reference for understanding potential therapeutic strategies that focus on PBMC metabolism, such as anti-inflammatory treatments.
General comment: Check the resolution of the images. It could be the copy, but the figures seem to be at a lower resolution and pixelated.
Minor comments:
Clarify how the sample size of 7 severe patients and 6 healthy humans were decided upon. Are there greater sample sizes to collate, or a particular reason to assume that it is normalized across demographics and age? Are there certain populations that or more or less susceptible? mitigating factors?
Are there more references to other metabolic studies to assist to determine what changes are statistically relevant? For a standard viral infection or other diseases, what would the flux be? Anything unique to SARS-CoV? or can be generalized.
Expand on the markers standardly focused on for anti-inflammatory treatments. Further search of what targets are being looked at for treatments and based on the DEG analysis clarify viability as targets for future development.
Table 1 and 2 are key highlights showing the top 25 metabolic reactions. A query how was it decided to show the top 25 versus 10 or 50, etc...?
Author Response
Response to Reviewer 1 Comments
General comment: Check the resolution of the images. It could be the copy, but the figures seem to be at a lower resolution and pixelated.
Response: According to the requirements of editor, the original images (> 300 bpi) were additionally packaged in "Figure. zip" to prevent the manuscript file containing the full images from being too large.
Point 1: Clarify how the sample size of 7 severe patients and 6 healthy humans were decided upon. Are there greater sample sizes to collate, or a particular reason to assume that it is normalized across demographics and age? Are there certain populations that or more or less susceptible? mitigating factors?
Response 1:
1) & 2) The donors donating blood samples for experimental group and control group were distributed in wide age spans (20-80 and 30-50, respectively), and the severe patients showed similar infection symptoms (admission level of 6/7 patients: ICU). So, these donors can be considered to be the representative of a certain community population, although older patients (one patients in the experimental group) have susceptibility.
3) Three patients received azithromycin for immunomodulation before sampling. Five patients were treated with remdesivir after admission. Four patients were treated with remdesivir before being sampled. (Wilk et al. 2020. Nat Med 26(7):1070-1076.)
As suggested, the above content has been added to the section “3. Materials and Methods”. (Lines 490-494)
Point 2: Are there more references to other metabolic studies to assist to determine what changes are statistically relevant? For a standard viral infection or other diseases, what would the flux be? Anything unique to SARS-CoV? or can be generalized.
Response 2:
1) The plasma metabolomics results of glutamine from other references were supplemented to support our predictions. The following has been added to the section “2. Results and Discussion”. (Lines 310-312)
“The exhaustion of glutamine in plasma as a metabolomic feature during COVID-19 infection indicates that adaptive immune cells in peripheral blood may increase the uptake of glutamine (Lee et al. 2022. Nat Biotechnol. 40(1):110-120.).”
2) Thank you for this valuable advice. As suggested, we’ve attempted to compare the differences of transcriptomics or metabolomics from PBMCs of patients infected with SARS-CoV (standard virus), MERS-CoV and SARS-CoV-2. Due to the unavailability of experimental data of PBMCs in SARS-CoV and MERS-CoV infection, the comparative of metabolic difference in PBMCs caused by the three coronavirus infections respectively can not be performed. We will continue to follow the availability of relevant data and carry out research on the metabolic differences of PBMCs induced by three viruses respectively in the future.
Point 3: Expand on the markers standardly focused on for anti-inflammatory treatments. Further search of what targets are being looked at for treatments and based on the DEG analysis clarify viability as targets for future development.
Response 3:
As suggested, we investigated and discussed the potential of metabolic enzymes responsible for responses with differential fluxes as anti-inflammatory markers. The following has been added to the “2. Results and Discussion”. (Lines 472-483)
“Furthermore, investigating the expression trend of metabolic enzymes responsible for differential flux can contribute to determine whether these active members possess the potential to become biomarkers of PBMCs for anti-inflammatory treatment. There were 10 up-regulated enzymes responsible for the increased flux of related metabolic reactions, including PKM, CMPK2, GAPDH, SLC38A5, SLC16A1, SLC2A3, GSR, MPO, CYP1B1, and GLB1. As a known metabolic marker in serum, GSR shows a significant negative correlation with increased IL-10 in COVID-19 patients (Naghashpour et al. 2022. J Med Virol. 94(4):1457-1464). In addition, MPO involved in steroid metabolism is considered as a neutrophil-derived infection marker to activate the production of cytokines and ROS in severe SARS-CoV-2 infection (Goud et al. 2021. Int J Biol Sci. 1;17(1):62-72). It is necessary to further investigate the abundance of metabolic enzyme in peripheral circulation or PBMCs, which can be applied for correlation analysis with cytokine level to support supervising infection process and assessing anti-inflammatory effect.”
Point 4: Table 1 and 2 are key highlights showing the top 25 metabolic reactions. A query how was it decided to show the top 25 versus 10 or 50, etc...?
Response 4:
Top 25 members can appropriately include the diversity of subsystems of metabolic reactions with the most obvious differential fluxes.
Reviewer 2 Report
In the submitted manuscript (ms), “Reconstructed genome-scale metabolic model characterizes 2 adaptive metabolic flux changes of peripheral blood mononu-3 clear cells in severe COVID-19 patients” authors have performed extensive computational analyses for establishing the relationship between peripheral immune cells and the pathogenicity arising by COVID-19 and potential drug target. It is well written & methods are described in detail. Herein only SRAS-CoV-2 taken under study, whereas, its variants where missing.
Author Response
Response to Reviewer 2 Comments
Response: We thank you for your recognition of our work. We will continue to follow the availability of experimental data on SARS-CoV-2 variants and research the metabolic disturbance in PBMCs induced by variants infection in the future.
Reviewer 3 Report
First i would like to compliment the authors for the time and effort invested in the protocol and manuscript writing.
The authors used COVID-19 patientes peripheral blood mononuclear cells in order to reconstruct a genome scale metabolic model.
Overall, the authors manage to communicate the main findings and ideas of the manuscript in a clear and understandable language. The transcriptomic data and metabolic network analysis is done properly and very detailed, using R and Python was a clever idea to analyze data. The authors are well aware about the limitations of their study.
Therefore, after carefully reading the entire manuscript I approve it for publication as it complies with all the guidelines of the journal and contributes positively to the global literature on COVID-19.
Author Response
Response to Reviewer 3 Comments
Response: We thank you for the positive comments.